# Contactless Gait Assessment in Home-like Environments

**DOI:** 10.3390/s21186205

**Published:** 2021-09-16

**Authors:** Angela Botros, Nathan Gyger, Narayan Schütz, Michael Single, Tobias Nef, Stephan M. Gerber

**Affiliations:** Gerontechnology & Rehabilitation Group, University of Bern, 3008 Bern, Switzerland; angela.botros@artorg.unibe.ch (A.B.); nathan.gyger@protonmail.com (N.G.); narayan.schuetz@artorg.unibe.ch (N.S.); michael.single@artorg.unibe.ch (M.S.); tobias.nef@artorg.unibe.ch (T.N.)

**Keywords:** gait analysis, gait abnormalities, contactless sensors, LiDAR, home-based measurements, health monitoring

## Abstract

Gait analysis is an important part of assessments for a variety of health conditions, specifically neurodegenerative diseases. Currently, most methods for gait assessment are based on manual scoring of certain tasks or restrictive technologies. We present an unobtrusive sensor system based on light detection and ranging sensor technology for use in home-like environments. In our evaluation, we compared six different gait parameters, based on recordings from 25 different people performing eight different walks each, resulting in 200 unique measurements. We compared the proposed sensor system against two state-of-the art technologies, a pressure mat and a set of inertial measurement unit sensors. In addition to test usability and long-term measurement, multi-hour recordings were conducted. Our evaluation showed very high correlation (r>0.95) with the gold standards across all assessed gait parameters except for cycle time (r=0.91). Similarly, the coefficient of determination was high ($R^{2}>0.9$) for all gait parameters except cycle time. The highest correlation was achieved for stride length and velocity ($r\geq 0.98,\;R^{2}\geq 0.95$). Furthermore, the multi-hour recordings did not show the systematic drift of measurements over time. Overall, the unobtrusive gait measurement system allows for contactless, highly accurate long- and short-term assessments of gait in home-like environments.

## 1. Introduction

Around one third of community-dwelling older adults are affected by gait abnormalities [1], which are associated with a variety of conditions that are commonly related to a reduction in quality of life. Respective conditions include, but are not limited to, neurological diseases such as Parkinson’s disease, Alzheimer’s disease and multiple sclerosis [2,3,4]. Commonly measured gait parameters are gait velocity, step length and cadence, which often serve as diagnostic and prognostic variables in medical assessments [4,5,6,7,8]. Most of the well-established clinical gait assessments are based on a set of predetermined tasks, which are manually scored by medical professionals. Prominent examples are: the timed Up and Go test (TUG) [9], the Berg Balance Scale (BBS) [10] and and the Functional Gait Assessment (FGA) [11]).

However, in recent years, technology-supported gait assessments have been gaining interest. There are several advantages of technology-supported measurements over traditional assessments. Most importantly, automated-technology-assisted assessments can improve the repeatability and objectiveness of assessments, thus significantly reducing inter-rater variability, which may eventually lead to more appropriate treatment or disease management options.

Among current gold standards to assess gait are pressure mats or walkways [12,13]. They are contactless and unobtrusive; however, they are restrictive in their use, in that they only allow for straight walks [14], require a physical installation and there is a small but non-zero risk of tripping. Currently, pressure mats are dominantly used in clinical settings under the supervision of medical professionals [15,16]. Another way to assess gait parameters is via motion analysis on the basis of motion capturing systems [17]. While they allow for high accuracy and high levels of detail concerning positions of body parts [18], they require the placement of visual trackers on the measured person’s body, as well as a large setup system, including multiple cameras [19]. Thus, both walkways and motion capturing systems are only suitable for measurements in clinical settings or other controlled environments. A less restrictive option are wearable sensors [20] for gait tracking [21,22].

Besides the usually short (less than a day) state of the art gait assessment in clinical settings, the need for long-term measurements has become more apparent. Its validity has been shown in multiple studies where long-term measurements have been implemented in various settings using ambient, object and wearable sensors [23,24]. Long-term measurements and monitoring allows people to stay in their own homes for longer before being admitted to an assisted living facility [25,26] or provide support for the personnel at assisted living facilities [27]. Long-term measurements assessed by sensor technology allows for the continuous evaluation of gait parameters over days and weeks, or for repeated measurements at higher frequencies.

An important point regarding why measurements at home are so important was made by Van Ancum et al. [28]. They showed that the velocity of natural gait, observed in everyday life, often follows bimodal distributions, one mode for slow and short walks, one mode for fast walk. In contrast, gait velocities obtained from measurements in clinical settings are not well represented by either of the two modes. Monitoring in everyday life also allows for screening of rare but critical events. A prominent example is falls, which occur in around 30% of the population over the age of 60 [29], and freezing of gait (FoG), a gait abnormality occurring in Parkinson’s disease (PD) patients [30]. Overall, knowing about these events, how often they happen, how they happen and what they look like can help improve the diagnostics, therapies and general knowledge about the underlying disorders [31,32,33]. For these reasons, clinical studies are increasingly including long-term measurements of gait, where people are measured over weeks and months to capture as many critical events as possible [34,35,36,37,38].

Another important aspect of gait assessment is the ease of use. While wearable sensors nowadays are small and lightweight, they can still impose a burden on the measured person, as they need to be worn and charged regularly. This burden can range from small issues, such as style disagreements or the stigmata of visible sensor wearing, up to more severe problems, such as skin irritations due to excessive sweating or bio compatibility [39]. Furthermore, people with, e.g., dementia or similar diseases might not remember to put on the wearable, lose them or actively get rid of them [32]. A contactless system is more or less immune to such issues.

Therefore, there is a strong need for a contactless system that is able to monitor gait over extended periods of time in a natural environment (e.g. at home), without putting a burden on the monitored people.

A promising technology is light detection and ranging (LiDAR) sensors. This technology has been used for object tracking in a variety of applications such as self-driving cars or geology [40,41]. This laser-based radar technology is both contactless and can cover whole rooms without restricting movement patterns to a fixed zone, and should be able to track multiple people accurately in a natural environment [42]. Galai et al. used a rotating multibeam (RMB) lidar to track and identify multiple people in a natural environment as well as to identify activities [43,44]. LiDAR sensors were used to track gait parameters in multiple studies [45,46,47,48]. Gait velocity, step and stride length, cadence and step width were the most commonly measured gait parameters. These past gait-related research projects have a commonality in that the LiDAR-based measurements were performed in a clinical setting, comparable to the setting where gait analysis is performed with a pressure mat. This is opposed to recommendations by Van Ancum et al., where performing gait assessment in home-like environments was suggested [28].

In this work, we introduce a new LiDAR-based contactless sensor system suitable for long-term measurements in everyday living settings in home-like environments. Our hypothesis is that with this proposed contactless system, it is possible to obtain highly competitive results with regard to gait parameter estimation in both home-like conditions as well as in controlled walking scenarios, showing that such systems may be a promising unobtrusive approach towards continuous gait parameter monitoring in free-living conditions.

## 2. Materials and Methods

### 2.1. Subjects

The study was approved by the Ethics Committee of the Canton of Bern, Switzerland (KEK-Nr. 406/16) and conducted in accordance with the latest version of the Declaration of Helsinki. The obtained data may be shared upon request, but is not publicly available due to local Swiss data regulations. The inclusion criteria for study participants were: (1) aged between 18 and 65; (2) the absence of a walking impairment affecting activities of daily living.

The appropriate number of participants was estimated beforehand, based on the planned number of different walks, the desired significance level α=0.01, power 1−β=0.99 and the effect size. The minimum number of different walks is seven; they are described in more detail in Section 2.4. We considered the two legs separately, resulting in 2·7·N sample points, with *N* being the number of participants. The effect size of the difference between our LiDAR-based system and any system currently already in use must not be higher than the effect size of the differences of the measured gait parameters between healthy control and people suffering from gait issues. Reference effect sizes were based on findings in the literature, both for PD and fall risk assessment [7,8,49,50]. An effect size of 0.3 was set for this study. The calculations for the necessary sample size were carried out using G*Power [51], and a minimum number of N=20 participants was calculated.

Both the experimental setup and the study purpose were described to the participants, and informed consent was obtained prior to participation. The study was conducted in a home-like instrumented apartment, the NeuroTec Loft [52] at the Swiss Institute for Translational and Entrepreneurial Medicine. In Figure 1, the layout of the living room and kitchen of the apartment is depicted.

This cross-sectional study was conducted over a period of three weeks. Every participant partook once in the experiment.

### 2.2. Sensors Used to Measure Gait

Pressure-sensitive mat:A pressure-sensitive mat (GAITRite^®^, USA [14]) was used as a reference system (gold standard) [53,54]. The GAITRite^®^ model *Platinum Plus Classic* had an active length of 610 cm and analysis was performed with the GAITRite^®^ software version 4.89F. The built-in metronome was used as an auditory cue to control walking cadence. The grey electronic-hiding cells on the side of the mat provided visual cues to control step length. The mat was installed in the living room of the apartment (see Figure 1). Gait parameters obtained from the pressure mat and used for evaluation were the step length, step time, stride length, cycle time, overall cadence and mean velocity.

Inertial Measurement Unit (IMU): A pair of validated IMU sensors (Physilog^®^5, Gait Up, Switzerland [20]) was used as a second reference system. One sensor was attached to each foot and measured acceleration and orientation data with a sampling rate of 128 Hz. The *Gait Up LAB* software (Gait Up, Switzerland) version 1.0.1 was used to perform the gait analysis. In order to ensure uniform sensor attachment, slippers of appropriate size were provided to the participants. Gait parameters obtained from the wearable sensors and used for evaluation were step length, stride length and cycle time. Additionally, this device provides left and right leg velocity and left and right leg cadence. These gait parameters were not used in the evaluation, as they differ from the gait parameters provided by the pressure mat.

LiDAR system: Four LiDARs (UST-10LX-H01 and UST-20LX-H01, Hokuyo, Japan) were installed in the living room (see Figure 1) of the Neurotec Loft.

LiDAR sensors use time-of-flight technology to measure distances using an infrared laser that rotates continuously around a mechanical axis. The LiDAR sensors used in this study scanned with a 40 Hz rate, acquiring 2162 points per turn. Every turn covered a 270${}^{\circ }$ window, leading to an angular resolution of $0.125^{\circ }$. Assuming a leg with a diameter of 10 cm, at a distance of 1 m from the LiDAR, around 45 sample points would cover the leg. At a distance of 10 m from the lidar, four to five points would still cover the leg.

All four LiDARs were connected to a local subnet using the ethernet protocol, automatically transmitting the data. The three UST-20LX-H01 sensors were positioned at a height of 25 cm to track both legs at shin height. An additional UST-10LX-H01 sensor was positioned at a height of 125 cm to track the upper body. The shin height was defined as approximately half the mean sitting knee height of 53.5 cm reported in [55] and is well-above the mean foot clearance height of 2 cm (SD 1 cm) reported in [56]. The height of the sensor placement is also in accordance with similar studies, where the LiDAR sensors were placed between 20 cm to 40 cm height [45,46,47,48]. The upper-body height was chosen so as to be above most of the furniture in the NeuroTec Loft, hence providing a direct line of sight throughout the room. A custom-made software acquired and analyzed the data.

### 2.3. Gait Parameters

The parameters chosen for evaluation were based on two criteria. First, their relevance in the evaluation of pathological gait behavior, and second, their current availability in clinical evaluations. For the first point, relevant gait parameters were evaluated based on quantitative assessment in previous studies. In Table 1, a selection of publications of relevant gait parameters for fall risk assessment, as well as PD assessment, are listed. All these parameters are provided by the pressure mat chosen for the gait evaluation [14], thus fulfilling the second criteria, too.

### 2.4. Study Protocol

After the instructions and having given informed consent, the participants put on slippers with the wearable sensors attached to them. Afterwards, participants were asked to perform a set of different walks. For seven of the eight walks, the participants had to walk straight along the pressure mat. For the eighth walk, the u-turn walk, the participants had to walk around the marker in the room and back to their starting location. In Figure 1, the marker is highlighted by a green triangle. The chosen walks were adapted from the Canadian Consortium on Neurodegeneration in Aging (CCNA) gait assessment tool [61], including walking at a self-chosen speed, fast-paced gait and dual-task gait, i.e., walking while performing a cognitively demanding task. For all eight walks, the starting position was the same at the left side end of the pressure mat. An overview of all the different walks is given in Table 2. In order to guarantee that the participants understood the experiment, they were able to do a test run prior to each walk.

For the first three walks, the participants were able to choose their own preferred walking speed and step length. For the first walk, there were no restrictions on the participants’ walking behavior, speed or step length, and thus it was a completely free walk in line with other gait assessment studies [45,46,47,48]. During the second walk, the participants had to perform mental calculations while walking. They were asked to count down from 100 in steps of 7. For the third walk, the participants were asked to walk with an extra wide step width, mimicking a cowboy-walk. Due to the extra wide stance, this walk was deemed to be simpler to measure. It was included in the task to provide a different stepping pattern, in order to test the limitations of our LiDAR-based sensor system. For the next four walks, a metronome was used to indicate the step timing. Marks on the floor were used to indicate step length. A short step length and slower gait velocity are more common in elderly people [62] and are an important indicator for fall risk assessment [6]. Thus, testing shorter steps with two different velocities was added to the protocol and in the fourth and fifth conditions, participants were asked to do 30 cm long steps at a cadence of 100 steps/min and 120 steps/min, respectively. In the sixth and seventh conditions, the step length was constrained to 60 cm, with a cadence of 100 steps/min and 60 steps/min. A longer step length was chosen, while the step velocity was further reduced to cover different conditions. For the last walk, a marker was placed in the middle of the room, near the couch. In Figure 1, it is highlighted with a green triangle. The participants were instructed to walk around this marker and back to the starting position. The speed and step length were not restricted. This walk was specifically added to test unrestricted walking in a large part of the living room, including obstacles (e.g., chairs, table).

### 2.5. Long-Term Functionality

The LiDAR sensor system is intended for unobtrusive long-term measurements of gait in home-like environments. To ensure long-term functionality and rule out drifting behavior of sensor data, a long-term measurement was set up. In the middle of the room, visible for all three LiDAR sensors, two pipe tubes were placed to mimic two legs. The pipes were placed such that all three sensors had a view of both legs and the algorithm should be able to discriminate between the two point clouds. The diameter of the pipes was selected such that it was similar to regular legs in tight-fitting trousers. Above the pipes, at upper-body level, a cardboard box was placed to mimic an upper body. The width of the box was similar to the upper torso of a healthy adult. The flat surface of the box was directed towards the UST-10LX-H01 LiDAR, placed at upper-body level. The LiDAR sensors were left to run continuously for 12 h and the obtained data were processed, as with the walking trials. The beginning and end of the data were trimmed to eliminate movement artifacts. However, to analyze individual sensor behavior, the leg positions were calculated for every sensor individually, instead of pooling the data together and obtaining more stable estimates. In the same manner, the *x*- and *y*-coordinates of the location information were considered individually. Location estimates were pooled together for every 15 min window, and for each window, median and quartile values were extracted.

### 2.6. Data Processing

The software described in this section is publicly available [63]. An overview of the data processing is given in Figure 2.

For the leg tracking process, there were always two data sequences acquired: a background sequence used for background removal (e.g., chairs, table) and the actual walking sequence for gait tracking. The background sequence was usually acquired ahead of the gait measurements when the room was empty.

Background Removal: The background sequences contained 2000 samples, equivalent to 50 s of recordings. The background removal step removed all static information of the living room, i.e., the background (e.g., chairs, table), such that only moving objects were remaining. This was accomplished by using a parametric approach, in which the probability distribution of every radial point was modeled with Gaussian distributions. For all points acquired, the probability of them belonging to the background was calculated. All points with a probability p<0.00001 were marked as foreground, while all other points were removed, as they belonged to the background. This method is similar to the approach by Frenken et al. [45].

Multi-Sensor Registration: The multi-sensor registration step was performed to align data from all four LiDAR sensors spatially and temporally. The spatial registration used a rigid transformation to align all coordinate systems. The rigid transformation was calculated beforehand, due to the fixed positions of the sensors. The time registration grouped the frames with the closest associated timestamps in all sensors together. After registration, data from all shin-level sensors were overlaid to form a single sequence of shin scans.

Clustering: A threshold-based breakpoint detection algorithm was used on the data obtained from the single upper-body sensor. Clusters were identified, and small clusters (n<5 points) were removed. The remaining largest cluster was determined to be the upper body. Its centroid was computed to estimate the center of the body. This upper-body estimate was used to filter out noise in the registered data from the leg-tracking sensors. Points farther away than a 50 cm radius from the upper-body centroid were considered noise and thus removed. The same breakpoint detection algorithm was applied to the remaining data from the three leg-tracking sensors. Small clusters (n<5 points) were removed. If, after the smaller clusters were removed, two clusters remained, the k-means algorithm was applied to obtain a cleaner clustering. The cluster centroid was computed to estimate the true shin position. If only one cluster was left, its centroid was directly assessed, and an estimate of the other missing leg position was computed using a particle filter. If no clusters were left, both leg positions were estimated using a particle filter [64]. A sequential importance resampling (SIR)-based particle filter [65] was used to estimate the missing information. The particle propagation was based on a linear velocity model, where the velocity corresponded to the upper-body velocity in the previous frame multiplied by a constant in the range [0.5,2.0].

Tracking: Leg tracking was performed to differentiate between the left leg and the right leg in each scan. The initial assignment was carried out by comparing the angle between the direction vector and the leg position vector referenced to the body center. For each shin position in subsequent frames, the same label was assigned as the nearest identified shin position in the previous frame.

Gait Analysis: In the gait analysis step, the identified shin positions and their associated timestamps were used to compute individual leg velocity profiles. A moving average filter was used to compute smooth velocity profiles. The stance point was set at the point with minimal velocity, the maximal swing point was the point with maximal velocity. Based on this information, the step and stride length, step and cycle time and cadence and overall velocity were computed. In Figure 3, the gait parameters are visualized.

### 2.7. Statistics

The developed LiDAR-based measurement system was compared against the gold-standard pressure mat and wearable IMU sensors. For all comparisons between the LiDAR-based measurement system and the pressure mat, only the seven walks performed on the pressure mat were used for the evaluation, i.e., all walks except for the u-turn walk. All eight walks were used for the comparison of the LiDAR sensors and the wearable sensors. As the wearable sensors computed slightly different gait parameters than the LiDAR-based system and the pressure mat, only matching gait parameters were compared (e.g., step length, stride length and cycle time).

For all matching gait parameters, two-sided, paired two sample *t*-tests were performed. We hypothesized that the gait parameter mean between each sensor pair was the same with α=0.01. Bonferroni correction was performed and adjusted *p*-values were reported, as well as the mean of differences and the confidence intervals. The $H_{0}$ hypothesis was that the two measurements were the same, and the $H_{1}$ hypothesis was that the two measurements differed in their mean.

To check general correlation between the measurements, the Pearson correlation, *r*, and the coefficient of determination, $R^{2}$, were calculated for every gait parameter and every sensor pair. Additionally, we calculated the intraclass correlation coefficient based on the two-way agreement mixed effects model (ICC3) [66]. The root-mean-square errors (RMSE) between the LiDAR based-measurements and the two baseline systems were calculated. For error distribution, Bland–Altman plots were generated and inspected; results are shown in the Appendix B.

## 3. Results

### 3.1. Demographics and Assessed Walks

A total of 25 (8 female, 17 male) healthy younger and older adults between the ages of 18 and 46 (mean age 30.5, SD 5.6) participated in this study. None of the subjects required any walking aids in everyday life.

In our measurements, through a free walk at a self-selected speed, as well as walks at a predefined speed, eight different walks were performed by every participant. These covered a wide range of different gait clusters, as shown in Figure 4. The walks with a predefined speed or step width formed tighter clusters of points, whereas the walks at self selected speed spread over a larger range of values. The walking velocities ranged from 40 cm/s up to 130 cm/s, while the stride lengths ranged from 50 cm up to around 160 cm.

### 3.2. Comparison of Devices

The results from the paired sample *t*-tests are summarized in Table 3. In the first block, the LiDAR sensors are compared against the pressure mat (PM). In the second block, the LiDAR sensors are compared against the wearable sensors (WS). In the last block, the wearable sensors are compared against the pressure mat. The adjusted *p*-value is specifically marked if it fell below α=0.01, meaning the means of the compared sets were different. There was a significant difference between the LiDAR and the pressure mat for stride length, with p=0.01. Furthermore, for this gait parameter, there was a significant difference in means between the pressure mat and the wearable sensors; meanwhile, there was no significant difference between the LiDAR and the wearable sensors. For one other gait parameter, cadence, the difference of means between the pressure mat and the LiDAR sensor was significant, too. The RMSEs were slightly higher between the LiDAR measurements and the wearable sensor, as compared to the pressure mat. Analysis of the Bland–Altman plots in Figure A2 shows no severe asymmetry or skewed distribution.

Additionally, there were high correlations (r≥0.95) between the LiDAR and the pressure mat across all gait parameters. When comparing the LiDAR sensors and the wearable sensors, the cycle time had the lowest correlation (r=0.91). The same holds for the comparison of wearable sensors and the pressure mat, where the cycle time had the lowest Pearson correlation (r=0.94). The coefficient of determination was overall high and only in two cases below 0.90—in both cases, the cycle time was the affected gait parameter. The Pearson correlation and coefficient of determination are presented in Table 4, whereas visualizations of all correlations are shown in the Appendix A in Figure A1. The overall agreement ICC3 between all three sensors was very high (ICC3≥0.94) for all gait parameters, except for cadence. For cadence, the consistency ICC3 was 0.95.

Besides the evaluated gait parameters, the LiDAR-based sensor system provided insights into detailed velocity profiles, even for walking patterns that were not along a linear path, as shown in Figure 5. In Figure 5a, an example of the u-turn walk is shown. Every point in this Figure relates to the centroid of a point cloud, as recorded by the LiDAR sensors, thus approximating the center of the corresponding leg. In Figure 5b, the smoothed velocity profile of a walking segment is shown. The colors for left and right leg in Figure 5a,b correspond.

### 3.3. Long-Term Measurement

The sampled location information over 12 h of measurement was evaluated. The median position is relatively constant and there is no significant change over time, as shown in Figure 6. In Figure 6, the location estimates are shown over time. The *y*-axis is scaled to cover 2.5 cm in all figures, but no absolute positional information is given. The upper-body location is shown in Figure 6a in *x*- and *y*-coordinates in the top and bottom picture, respectively. The *x*- and *y*-coordinates of the leg positions, as estimated by the LiDAR sensors placed at shin level, are shown in Figure 6b–d.

## 4. Discussion

In this work, we presented a LiDAR-based sensor system, capable of tracking clinically relevant gait parameters such as gait velocity and step length in a home-like environment in healthy subjects. Our results indicate that the proposed unobtrusive LiDAR-based gait tracking system allows one to accurately assess gait parameters, as is evident by the high to very high correlation with both comparison systems. Furthermore, the LiDAR-based system was able to generate detailed velocity profiles for each leg. Long-term measurements over more than 11 h did not give any significant indication that the measurement drifts over time.

The general agreement between the three sensors was very high, except for stride length and cadence. It is noteworthy that in the case of stride length, the comparison of the LiDAR and the wearable sensor showed no significant difference in means, while there was a significant difference in means between the wearable sensor and the pressure mat. Nevertheless, in both cases, the Pearson correlation was very high and suggests that different methods of calculation might have led to this discrepancy. Furthermore, the differing offsets and/or gradients can be corrected for easily, to align the results, if desired. For all compared gait parameters, the coefficient of determination was high, except for two cases: when comparing the cycle time of the wearable sensor with any of the two other sensors. This was caused by an outlier on the data obtained from the wearable sensors.

The second main finding indicates that our sensor system is suitable for long-term measurements in unrestricted, home-like environments. The measurement did not detect any significant drift in measurement, and the assessed leg and body positions were constant and thus reliable over time, indicating that the system was in line with previous findings [54].

Based on these two findings, we infer that long-term measurements in natural settings are feasible. This is especially important with regards to the findings by Van Ancum et al. [28], that gait speed measured in clinical settings is not fully representative of natural gait. Considering that gait parameters are often associated with and are important for the assessment of geriatric diseases such as Parkinson’s disease, a gait monitoring system installed at home or in assisted living facilities would cover a significant portion of the day, and thus reflect the optimal gait speed.

In comparison with the literature, the accuracy of the gait parameter measurements was in line or even higher than previous studies using LiDAR systems [45,46,47,48]. In these past studies, gait velocity and step length were often the only assessed parameters, whereas we could show evidence that the system is able to measure a wide variety of gait parameters.

Whereas most studies used a marker based motion tracking system to validate their LiDAR based gait assessment, we chose to compare our measurements to the current gold standards used for clinical gait assessment [15]. The knowledge of how gait assessment is performed currently and of how gait parameters obtained from these gold standards transfer to the gait parameters obtained from our new system allows for the direct interpretation of the results.

### Limitations and Outlook

In this study, the system was evaluated with data from healthy subjects. Even though a variety of different gait patterns were tested, it is not clear to what extent these results can be transferred to patients with gait abnormalities. Therefore, as a next step, measurements with elderly people and people with gait abnormalities should be performed. These gait abnormalities should include limping, shuffling, freezing of gait and specifically small steps, as is typical for people at risk of fall [30,31,37].

Another major limitation of our system is that it only allows for a single person to be tracked. This means couples living together or visitors must be excluded in the existing system. While adults living alone are still a key target group, especially when it comes to falls [37], this certainly limits the range of potential applications. As has been shown in previous work [42,43], LiDAR systems can theoretically track multiple people, so this a key point for further development of the introduced system.

Another limitation of the proposed approach is the ability of the wearable sensor setup to assess single and double leg support, swing time and stance time [20]. However, through further exploitation of the detailed velocity profile and more integration of domain knowledge, it is plausible to assume that these parameters can be estimated with the proposed setup.

Another interesting parameter is the upper-body position, which is currently only used for position estimation of the legs. The upper-body measurements might allow one to extract information for posture assessment.

## 5. Conclusions

Consistent with our first hypothesis, our results indicate that the assessment of gait parameters in a home-like environment based on LiDAR technology correlated highly with the clinical gold-standard devices. Secondly, the LiDAR-based system was able to accurately measure the selected wide spectrum of gait patterns, representing gaits from a home-like environment, including a free walk through the room. Third, the intra-test reliability of the assessed gait parameters indicate that the measurements did not significantly drift or variate over time and allow for long-term measurements over days and weeks. Thus we conclude that due to its unobtrusiveness, the LiDAR system has high potential to assess spatial–temporal gait parameters over a prolonged time at patients’ homes or home-like environments, such as assisted living facilities.

## Figures and Tables

**Figure 1 sensors-21-06205-f001:**
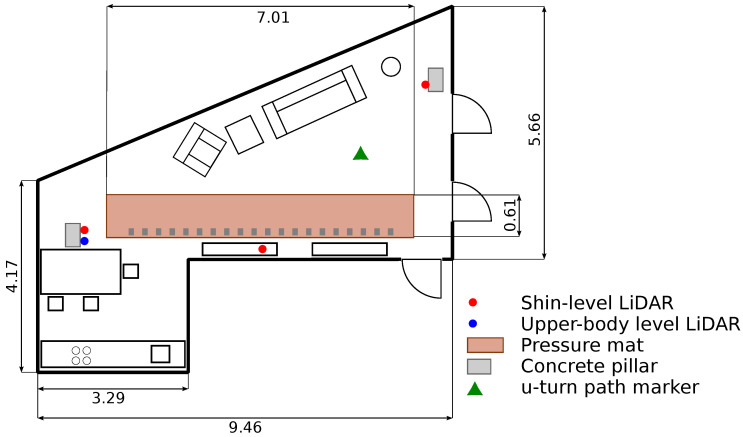
Schematic view of the living room and kitchen areas with the shin-tracking sensors (red), upper-body-tracking sensor (blue), marker for the u-turn walk (green), and pressure-sensitive mat (light red). All distance measurements are in meters (m). Sensor position 1 for shin-tracking is on the left on a pillar, sensor position 2 is in the middle on a side-board. Sensor position 3 is to the right on a pillar.

**Figure 2 sensors-21-06205-f002:**
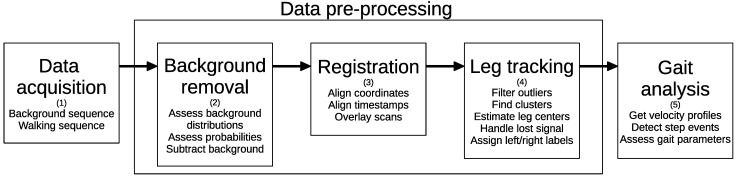
Data processing workflow. LiDAR data were first acquired (1), then the background data points were removed (2). Next, data from all sensors were aligned and overlaid using spatial as well as time registration (3). Then, clustering algorithms were applied to identify body clusters which were tracked across data frames (4). Finally, gait analysis was performed (5).

**Figure 3 sensors-21-06205-f003:**
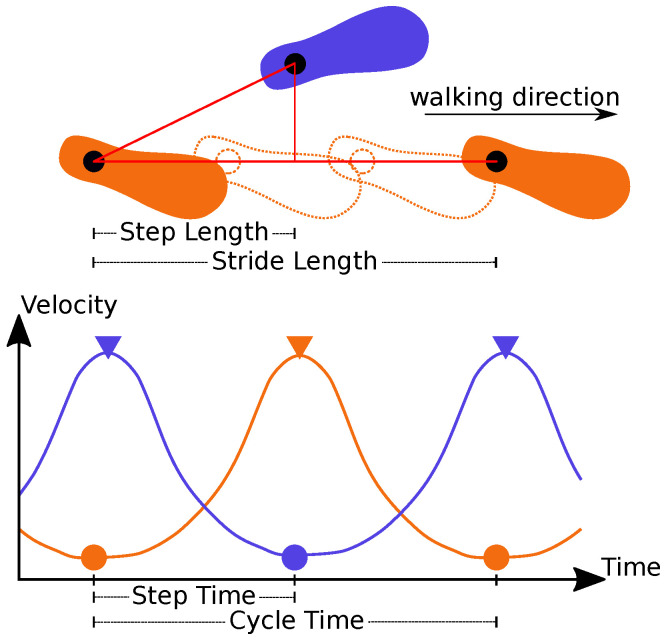
Calculation of gait parameters was based on the velocity profile of the tracked legs. Standing positions in the upper part of the figure are indicated at minimal velocity with circles in the lower part of the figure. The peaks of the swing phase are indicated with triangles.

**Figure 4 sensors-21-06205-f004:**
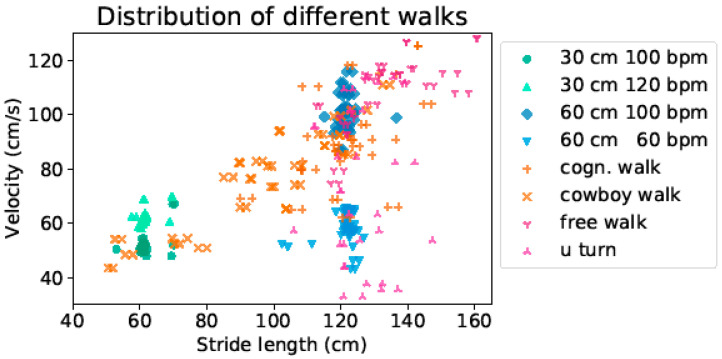
Green triangle circles in the left bottom corner mark the two walks with 30 cm step length at 100 bpm and 120 bpm, respectively. The blue diamonds and triangles on the right side mark the two walks with 60 cm step length at 60 bpm and 100 bpm, respectively. The orange and red markers indicate the four free walks, where participants were able to choose both their step length as well as walking speed individually.

**Figure 5 sensors-21-06205-f005:**
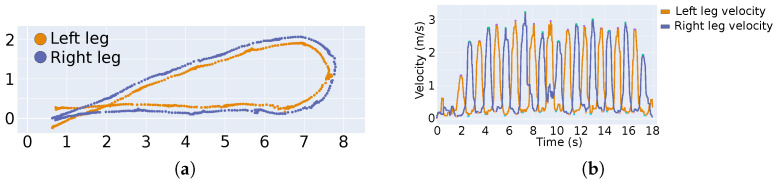
U-turn walk at self-selected speed and step length. (**a**) shows the identified leg positions during a walk. Denser areas correspond to stance phase, whereas scarcer areas correspond to swing phase. (**b**) shows the velocity profiles over time. Left and right leg are indicated by colors.

**Figure 6 sensors-21-06205-f006:**
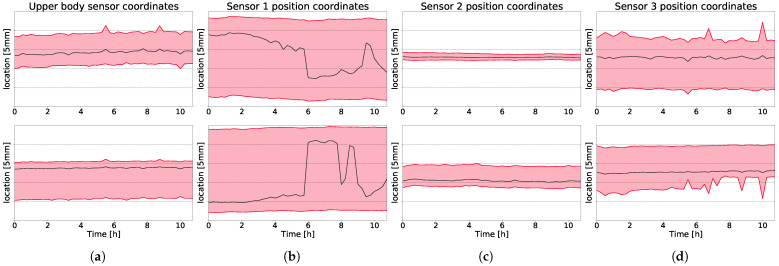
Long-term location for every LiDAR sensor split for *x*- and *y*-direction. In (**a**), data from upper body sensor is shown, its location is denoted with blue marker in Figure 1. In (**b**–**d**), data from the leg tracking sensor are shown. Their location is marked in Figure 1 with red markers. Top row shows *x*-position of upper body and legs, bottom row shows *y*-position of upper body and legs. The black line indicates the median value, the red-shaded area is the 2nd and 3rd quartile. The *y*-axis is scaled to 2.5 cm, with every tick indicating 5 mm. No absolute position units are given.

**Table 1 sensors-21-06205-t001:** Chosen gait parameters and evidence for quantitative relevance in clinical evaluations.

	Fall Risk	PD
Step length	[7]	[49]
Step time	[7]	-
Stride length	[7,57,58]	[8,49,50,59]
Cycle time	[7,58]	[49,50]
Velocity	[57,58]	[8,49,50,59]
Cadence	[58,60]	[49]

**Table 2 sensors-21-06205-t002:** Eight different walking patterns performed by the participants. Four walks were performed without fixed step length of time. For the other four walks, floor-markings and a metronome were used to control step length and walking speed.

	Velocity	Step Length	Defining Characteristics
Free walk	-	-	Self-chosen pace and step length
Cognitive walk	-	-	Cognitive exercise while walking
Cowboy walk	-	-	Extra wide cowboy-like steps
30/100	100 bpm	30 cm	Fixed step length and velocity
30/120	120 bpm	30 cm	Fixed step length and velocity
60/60	60 bpm	60 cm	Fixed step length and velocity
60/120	120 bpm	60 cm	Fixed step length and velocity
U-turn	-	-	Self-paced walk around a marker

**Table 3 sensors-21-06205-t003:** Descriptive statistics for the three sensor systems: LiDAR, pressure mat (PM) and wearable sensors (WS). Statistics were computed across sensors to test for differences in the sample mean. Not all gait parameters could be calculated for all systems. Unavailable gait parameters are indicated by a dash. The adjusted *p*-value is specifically marked if it fell at or below α=0.01.

		Step Length	Step Time	Stride Length	Cycle Time	Cadence	Velocity
		(cm)	(s)	(cm)	(s)	$\left(\frac{1}{\min}\right)$	$\left(\frac{\mathrm{cm}}{s}\right)$
LiDAR-PM	mean	−0.09	−0.00	−0.63	−0.00	8.52	−1.09
	conf. int.	[−0.45, 0.27]	[−0.01, 0.00]	[−1.04, −0.22]	[−0.01, 0.01]	[7.85, 9.18]	[−1.66, −0.52]
	adj. *p*-value	1	0.09	**0.01**	1	**<0.01**	0.04
	RMSE	3.31	0.05	3.84	0.09	10.48	5.33
LiDAR-WS	mean	−0.37	-	0.22	−0.00	-	-
	conf. int.	[−0.97, 0.23]	-	[−0.47, 0.91]	[−0.02, 0.01]	-	-
	adj. *p*-value	0.67	-	1	1	-	-
	RMSE	4.93	-	5.63	0.14	-	-
WS-PM	mean	−0.25	-	1.00	−0.00	-	-
	conf. int.	[−0.72, 0.22]	-	[0.52, 1.49]	[−0.014, 0.01]	-	-
	adj. *p*-value	0.74	-	**<0.01**	1	-	-

**Table 4 sensors-21-06205-t004:** Correlation comparison of LiDAR sensors, pressure mat (PM) and wearable sensors (WS). For all three pairs of measurements, the Pearson correlation (*r*) and the determination coefficient ($R^{2}$) are shown. The agreement ICC3 over all three sensors is shown in the last row. Dashes represent cases where a gait parameter could not be calculated by the respective sensor.

		Step Length	Step Time	Stride Length	Cycle Time	Velocity	Cadence
LiDAR-PM	*r*:	0.98	0.96	0.99	0.96	0.98	0.95
	$R^{2}$:	0.95	0.91	0.98	0.92	0.95	0.90
LiDAR-WS	*r*:	0.95	-	0.98	0.91	-	-
	$R^{2}$:	0.90	-	0.97	0.83 (0.91)	-	-
WS-PM	*r*:	0.97	-	0.99	0.94	-	-
	$R^{2}$:	0.94	-	0.98	0.89 (0.97)	-	-
LiDAR-WS-PM	ICC3:	0.96	0.95	0.99	0.94	0.98	0.86 (0.95)

## Data Availability

The data are not publicly available due to Swiss privacy regulations.

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
