# Peer review of "Contactless Gait Assessment in Home-like Environments"

_sensors, 2021, doi:10.3390/s21186205_

Round 1

Reviewer 1 Report

In this paper, the authors propose a LIDAR system to analyse gait of patients in their home environment. The system is compared with a wearable device and with a pressure mat through a paired t test and the correlation of Spearman. The paper might be interesting but there are some major concerns that must be addressed before the paper can be considered eligible for publication:

  1. The introduction is too long, maybe it could be synthetized and focused on the most important aspects.
  2. The references are too old. There have been new research papers in the last years regarding Parkinson and gait analysis (with and without machine learning). Please find some examples:
    1. Picillo M, Ricciardi C, Tepedino MF, Abate F, Cuoco S, Carotenuto I, Erro R, Ricciardelli G, Russo M, Cesarelli M, Barone P and Amboni M (2021) Gait Analysis in Progressive Supranuclear Palsy Phenotypes. Neurol. 12:674495. doi: 10.3389/fneur.2021.674495
    2. Donisi, L., Pagano, G., Cesarelli, G., Coccia, A., Amitrano, F., & D'Addio, G. (2021). Benchmarking between two wearable inertial systems for gait analysis based on a different sensor placement using several statistical approaches. Measurement, 173, 108642.
  3. Why did the authors choose only those six parameters?
  4. I’m not sure that putting the “materials and methods” section after result and discussion is a good choice. It tends to confuse the readers.
  5. Cadence, step time and velocity can usually be computed through IMUs. Why do the authors report that they cannot be computed?
  6. When using a t test for multiple comparisons, the correction of Bonferroni must be used. Moreover, since a comparison among three systems was performed, maybe using anova and then a post hoc t test would have been more correct.
  7. It is not clear to me why acquiring data for several hours should be difficult. Smartphones and smartwatches already acquire accelerometers, gyroscopic and magnetometer signals continuously over time without any issues. Which is the added value of LIDAR versus these devices?
  8. Is is not clear which is the real advantage of the proposed LIDAR system
  9. Why didn’t the authors consider comparing their system with a motion capture system which is among the most accurate ones (even if more complex)?
  10. The acronyms should be defined when first cited and then used consistently across the manuscript.

Reviewer 2 Report

Thank you for submitting this paper to Sensors. The manuscript under consideration: "Contactless Gait Assessment in Home-like Environments" is an interesting article on an important topic in Sensors. However, there are a few concerns.

1. In the study participants with age between 18 and 65 participated. How did the authors determine the sample appropriate size? 

2. Give more information on the age of the sample, why it was chosen, and give gait ability at this age.

3. Why did authors select age between 18 and 65? From the young and older adults, it is possible to obtain measurements of gait pattern on difference in age. Authors should clearly describe the purpose of selecting age between 18 and 65.

4. Please add more information about the research design (cross-sectional study) such as clarify period the researchers spent to collect data? Is it impact the cross-sectional data?

Reviewer 3 Report

The context of the study, i.e. countless gait assessment in home-like environments is significant for online monitoring of neurodegenerative patients. However, this paper should be extensively revised by enhancing its originalities and giving clearer expeimental analysis. Concretely, the advantage of using this new LiDAR system related to the existing work in a long duration homecare should be clearly presented in Introduction. Moreover, in the design of experiments, the subjects selected are mainly normal people without sympotoms. It ill be more interesting to introduce new subjects with mentioned patients and compare their results with normal people. In the analysis of results, only difference between the three devices has been compared but the accuracy, sensitivity, robustness and other performance indices have not been clearly compared and analyzed.  Also, the gait data analysis is relatively weak. The velocity profile should be merged with other measures to determine gait status (data fusion). The data filtering  and other processing steps should be more clearly analyzed.  Finally, for the proposed LiDAR system, its general principles and main components and their connection should be more clearly described, showing that it has more advantages than the other systems from theoretical point of view.

Round 2

Reviewer 1 Report

The authors have adequately addressed all the comments.

Reviewer 3 Report

The remarks of the review report in the 1st round have been fully taken into account in the new version. The paper can be considered for publication.